# Treatment of Severe Atopic Dermatitis with Dupilumab in Patients with Advanced Cancer

**DOI:** 10.3390/jcm12031191

**Published:** 2023-02-02

**Authors:** Milena Tanczosova, Jan Hugo, Spyridon Gkalpakiotis

**Affiliations:** Department of Dermatovenereology, Third Faculty of Medicine, Charles University and Kralovske Vinohrady University Hospital, 10034 Prague, Czech Republic

**Keywords:** atopic dermatitis, dupilumab, cancer, real-world study, biological therapy

## Abstract

Atopic dermatitis is a chronic inflammatory intensively pruritic skin disease. Patients with moderate-to-severe atopic dermatitis or with difficult-to-treat areas are candidates for systemic therapy, especially when topical therapy is inadequate. Currently, we have available not only conventional immunosuppressive systemic therapy, but also targeted biological therapy, which has shown a remarkable reduction in clinical severity with a good safety profile. Dupilumab has been approved to treat moderate-to-severe atopic dermatitis. Even though the therapy has been available for more than 3 years, there are still limited data regarding the treatment of patients with concomitant cancer. Previous immunosuppressive treatment for atopic dermatitis, such as cyclosporine or azathioprine, poses a safety risk for patients with malignant disease. We present a case series of three patients with advanced cancer and severe atopic dermatitis treated with dupilumab for an average of 17 months with a great response toward atopic dermatitis without cancer recurrence. One patient had colorectal cancer’ the second and the third both had cancer duplicity—colorectal and kidney cancer and penile squamous cell carcinoma with prostate cancer. Our cases suggest that dupilumab can safely control atopic dermatitis in patients with advanced cancer.

## 1. Introduction

Atopic dermatitis (AD) is one of the most common chronic inflammatory skin diseases of a noninfectious nature, which, although not life-threatening, has a significant negative impact on the patient’s quality of life. The prevalence of AD has doubled to tripled in industrialized countries since the 1970s, with approximately 15% to 20% of children (of whom up to 30% have moderate-to-severe AD) and 2% to 4% of adults (of whom up to 46% have moderate-to-severe AD) affected worldwide [1,2].

Although immune dysregulation and skin barrier defects are accepted as key components in the development of the disease, the pathophysiology remains unclear despite ongoing scientific research. While previous studies have largely focused on immune dysregulation of T helper cell type 1 (Th1) and Th2 groups as the pathogenesis of the disease in genetically predisposed individuals, skin barrier disruption and systemic inflammation are the focus of current AD research [3]. In AD, disruption of the skin barrier results in increased transepidermal water loss, decreased skin hydration, and increased presentation of antigens by Langerhans cells initiating inflammation [3].

However, two hypotheses are now known regarding the origin of inflammation that leads to the triggering of atopic dermatitis [4].

The first hypothesis is primarily an immune dysfunction (increased expression of Th2 lymphocytes and, thus, production of interleukins IL-4 and IL-13), resulting in IgE sensitization, allergic inflammation, and a secondary role of impaired epithelial function [4].

The second hypothesis is that a primary role in the development of atopic dermatitis is played by impaired epithelial function of the skin, leading to immunological dysregulation and subsequent inflammation. Genetic and environmental factors are also involved in the expression of atopic dermatitis [4].

As a result of these multiple factors, atopic dermatitis exhibits significant heterogeneity in disease phenotype, age of onset, clinical severity, persistence, comorbidity, and response to treatment. The clinical picture of atopic dermatitis, as well as the course of the disease itself, is diverse and unpredictable. The basic characteristic triad consists of persistent pruritus, dermatitis, and xerosis. Pruritus is a key and dominant feature of AD [5]. It generates comorbidities such as sleep loss and psychological distress, creating a continuous burden for patients, parents, and other family members [6]. Atopic dermatitis has a negative impact on patients’ normal daily activities [7] and their work productivity [8], and it can be accompanied by many comorbidities such as psychiatric diseases, ophthalmological disability, cardiovascular disease, anxiety, depression, and different autoimmune diseases [6].

The existing evidence on cancer risk in atopic dermatitis is inconsistent, with studies limited by insufficient consideration of severity of atopic dermatitis and its treatment. A recent population-based cohort study by Wan et al. reported a reduction in the incidence of certain solid tumors (e.g., breast cancer) and an increased risk of lymphoma in patients with atopic dermatitis [9].

Furthermore, atopic dermatitis is also associated with significant direct and indirect financial costs, which correlate with disease severity [10].

The goal of therapy is to achieve long-term remission with minimal side-effects and to improve patients’ quality of life. Options of topical therapy are limited to corticosteroids and topical immunomodulators [11]. Topical therapy is the mainstay of AD treatment and can be used in monotherapy for mild cases. In patients with severe atopic dermatitis, it serves as an adjunct to systemic therapy.

Systemic therapy, including biological, is indicated for patient with moderate-to-severe forms of atopic dermatitis. Among the conventional immunosuppressive drugs, cyclosporine is the drug of first choice in adult patients as it is the only on-label conventional systemic treatment in Europe [11,12]. Other immunosuppressants can also be used off-label, such as systemic corticosteroids, azathioprine, mycophenolate mofetil, methotrexate, or phototherapy [11]. Although these traditional immunosuppressive therapies can be effective in atopic dermatitis, their routine use is often limited due to adverse effects, frequent laboratory monitoring, and inappropriateness of long-term usage [12]. Furthermore, it is always necessary to consider comorbidities and contraindications, as well as possible secondary infections, and to exclude malignancy due to the immunosuppressive effect. With systemic therapy, it is important to carefully consider the risks and benefits of the selected drug. The newest treatment option for patients with atopic dermatitis is targeted therapy, the development of which has been made possible by a closer understanding of the pathogenesis of AD at the molecular level. Knowledge of the immunological background of AD has led to the synthesis and testing of monoclonal antibodies targeting these cytokines, with the first approved drug being dupilumab [12,13]. The second group of targeted therapies for AD constitutes small molecules whose mechanism of action is JAK/STAT inhibition [14].

Dupilumab is a fully human monoclonal antibody against the α chain of the interleukin (IL) 4 receptor; it blocks IL-4 and IL-13 signaling pathways and has shown a remarkable reduction in clinical severity with a good safety profile in both clinical trials and real-world studies [13,15,16,17,18]. IL-4 and IL-13 are key Th2 cytokines that play a critical role in the pathophysiology of allergic diseases and, thus, in the development of atopic eczema [4].

Dupilumab has been reimbursed in the Czech Republic since 1 June 2019 for adult patients with severe AD after failure or ineffectiveness of at least one conventional systemic immunosuppressive therapy (except systemic treatment with corticosteroids) or for patients in whom the use of conventional systemic therapy is contraindicated. Dupilumab is also reimbursed for patients aged 6–18 years after failure of maximal topical therapy and phototherapy or balneotherapy

No specific laboratory or imaging tests are required before or during therapy. Treatment of pre-existing infections is recommended [19]. The most common side-effects are application site reactions, conjunctivitis, blepharitis, and herpes simplex infection. Dupilumab can be combined with topical corticosteroids or topical immunomodulators.

Furthermore, the FDA has approved dupilumab for the treatment of bronchial asthma, chronic sinusitis with nasal polyposis, eosinophilic esophagitis, and prurigo nodularis [19].

Long-term safe therapy is the only effective way to prevent recurrence and exacerbation of AD.

Patients with a history of malignancy were generally excluded from dupilumab clinical trials [13,16]. Current conventional immunosuppressive treatment for AD such as cyclosporine or azathioprine is not suitable for patients with a history of cancer. Up to now, there are no data reporting that modulation of the IL-4 and IL-13 signaling pathways can increase the risk of malignancy [20].

Although interventional clinical trials have demonstrated the efficacy and safety of dupilumab, they may not accurately correspond to “real-life” practice, given that clinical trials often do not include patients with comorbidities (e.g., decompensated arterial hypertension, severe eye diseases, and patients with previous oncological disease), which we commonly see in clinical practice.

Only a few cases have been reported of patients with advanced cancer being treated with dupilumab [21,22]. Here, we present three patients with atopic dermatitis and previous malignancy treated with dupilumab, with good clinical response on AD and no evidence of cancer recurrence.

## 2. Case Series

Case 1. Our patient is a 59 year old nonsmoker female suffering from atopic dermatitis since the age of 40, which first appeared after significant psychological stress. The manifestations of atopic dermatitis were initially mild and then gradually proceeded to a generalized phenotype. She was treated with UVB phototherapy (311 nm) which triggered an exacerbation of skin lesions. The patient suffered from severe pruritus and inability to sleep. She had a personal history of arterial hypertension and hypothyroidism on pharmacological therapy. Of the atopic comorbidities, only polyvalent allergy was present. Because of severe atopic dermatitis, the patient was offered participation in a double-blind, randomized trial with nemolizumab; however, soon after the initiation, her participation had to be terminated because she was diagnosed with stage IIIb colorectal cancer. The patient underwent resection and subsequent adjuvant chemotherapy with capecitabine, which had to be terminated after the seventh cycle due to significant progression of atopic dermatitis. After consultation with the attending oncologist, the use of conventional systemic treatment (cyclosporine, methotrexate, and azathioprine) was contraindicated due to the induction of immunosuppression, which could accelerate the primary malignancy, especially due to the fact that capecitabine therapy was terminated prematurely and, thus, the risk of malignancy relapse was higher. We indicated the patient to use targeted biological treatment with dupilumab in a standard dosage regimen. At the beginning, she had an Eczema Assessment Severity Index (EASI) of 24.4, a Dermatology Life Quality Index (DLQI) of 19, and an itch Numerical Rating Scale (NRS) of 7; her serum IgE level was 1700 IU/mL. Ophthalmological atopic comorbidities were excluded after ophthalmological examination. The patient has now been on dupilumab therapy for more than 1 year. The EASI value is 0.9, which means that she is a EASI90 responder; the DLQI is 0, NRS is 2, and serum IgE level has decreased to 470 IU/mL. There have been no side-effects, and the patient is still in remission from the oncological disease.

Case 2. Our second patient is a 46 year old female with a history of atopic dermatitis, bronchial asthma, polyvalent allergy, and cancer duplicity—colorectal cancer stage IIIa and carcinoma of right kidney stage Ia, both treated with surgical removal, followed by adjuvant chemotherapy and radiotherapy, 2 years before she visited our department. Atopic dermatitis was under control with only occasional application of topical corticosteroids until she was diagnosed with cancer. After the cancer treatment, she presented to our department with severe pruritus NRS 6/10 and a diffuse eczematous eruption—EASI score of 50 (Figure 1). Impact on quality of life was severe with a DLQI score of 15; she complained of inability to sleep due to continuous pruritus, her work productivity was seriously impaired, and she had to seek psychological help.

Considering her oncologic anamnesis, we initially treated her with topical treatment in combination with narrowband UVB phototherapy. However, the condition did not improve. Systemic immunosuppressive therapy was considered unsafe due to the previous malignancies. After discussion with her oncologist, we initiated treatment with dupilumab in a standard dosage regimen. The patient responded very well to the treatment (Figure 2); she has been currently treated with dupilumab for 2.5 years, her EASI score is 8, her subjective pruritus has decreased to 1, and her DLQI score has improved to 3. The patient is very satisfied with the results of dupilumab treatment. To date of this publication, no adverse events have occurred. The patient is following up with her oncologist, and surveillance imaging are negative to date.

Case 3. A 56 year old man with a lifelong history of AD, bronchial asthma, and polyvalent allergy presented to our department with severe pruritus and generalized eczematous lesions, which did not respond to topical treatment or to application of systemic corticosteroids. He had a history of penile spinocellular carcinoma and prostate cancer that was diagnosed 2 years before our treatment initiation, both treated only surgically, in remission at the time of his visit. Pruritus and inability to sleep had a high impact on his quality of life. Owing to his positive cancer history, he was unable to participate in clinical trials since he would not fulfill the inclusion criteria.

We initiated narrowband UVB phototherapy in combination with topical corticosteroid therapy. However, atopic dermatitis did not improve, and he continued to experience frequent flares. Given the severity of his AD, the impact on his quality of life, and cancer history, systemic immunosuppressive drugs were contraindicated by his oncologist; hence, we decided to start biological treatment. Dupilumab was initiated at a 600 mg loading dose, followed by 300 mg biweekly in a standard dosage regimen. At the beginning of the treatment the EASI score was 32.9 (Figure 3), reported pruritus was 8/10, and DLQI score was 16; the IgE level was 1670 IU/mL. After 4 months of treatment, the EASI score dropped to 4, pruritus score decreased to 1, and DLQI score was 2 (Figure 4). Serum levels of IgE also decreased to 937 IU/mL. The patient continues therapy with dupilumab with good response and no side-effects. After 5 months on therapy, no recurrence of cancer was observed.

## 3. Discussion

Atopic dermatitis (AD) is one of the most common chronic inflammatory dermatoses of noninfectious nature, which, although not life-threatening, significantly affects the patient’s quality of life [5]. The treatment of atopic dermatitis has been for long time very complicated due to the fact that available drugs were not suitable for long-term administration, and their use was accompanied by side-effects and is inappropriate for patients with certain comorbidities such as immunodeficiency or malignancy.

The aim of therapy is to achieve remission with minimal drug toxicity and to improve the patient’s quality of life.

Significant improvements in the understanding of the etiopathogenesis of atopic dermatitis have been seen in the last few years.

A recent systematic review of preclinical and clinical studies showed that there is no increased risk of malignancy when specifically targeting IL-13 and IL-4 [20]. Data from registries have also not reported an increased risk of malignancy [23,24].

To date, few reports have described real-life experiences of the use of biological drugs in atopic dermatitis patients with a history of malignancy [21,22]. Fowler et al. described two patients with cancer (malignant melanoma and anal squamous cell carcinoma) who were treated with dupilumab safely with good clinical response on atopic dermatitis [21]. Another case report by Qiu et al. presented a patient with history of non-Hodgkin’s lymphoma (NHL) with multiple relapses and a severe form of atopic dermatitis. The patient responded to treatment with dupilumab very well, with no reported side-effects [22].

None of the published case reports showed any risk of cancer recurrence for patients with an oncologic history after dupilumab treatment [21,22].

On the other hand, a case of a bladder cancer during dupilumab therapy was recently described; although the authors did not associate the tumor with dupilumab treatment, they stopped the biologic therapy [25]. Interestingly dupilumab has been used in patients with multiple myeloma (MM) for the treatment of lenalidomide rashes. Multiple myeloma remained stable during dupilumab application suggesting that it could have therapeutic benefits as an adjuvant therapy for MM [26]. There is a clinical trial currently enrolling subjects to investigate the safety, antitumor effect, and immunogenicitiy of neoadjuvant dupilumab given prior to radical prostatectomy in men with high-risk localized prostate cancer [27] and another clinical trial that is enrolling subjects to investigate whether dupilumab may be beneficial in patients with metastatic non-small-cell lung cancer following immunotherapy [28]. There is an unproven suggestion that dupilumab may be beneficial for downregulating PD-1 expression on tumor-infiltrating lymphocytes and for improving cancer immunotherapy [28].

In a murine model, IL-4 blockade led to an increase in IL-12, IFNg, and TNF in CD8^+^ T cells and a reduction in tumor burden, and additional antitumor activity was observed in combination with PD-L1 blockade. On the basis of these preclinical data, it is hypothesized that the addition of dupilumab to anti-PD-(L)1 therapy will be well tolerated and will preserve the antitumor effect of immune checkpoint blockade [28].

We described three cases with severe atopic dermatitis unresponsive to topical therapy and to narrowband phototherapy with a history of malignancy and current dupilumab treatment (Table 1).

Our patients had a mean interval between cancer diagnosis and initiation of dupilumab treatment of 1.7 years. The tumor types differed from the previous published case series [21,22].

All our patients had a severe form of atopic dermatitis at the time of the visit to our department; they complained of severe pruritus, and their quality of life was also impaired. None of them had any previous systemic treatment for atopic dermatitis. Due to their history of malignant disease, we decided in accordance with their oncologists to not initiate conventional systemic treatment. On contrary, dupilumab therapy is not connected to a higher risk of malignancy.

We obtained IgE levels in the patient with colorectal carcinoma, as well as in the patient with penile spinocellular carcinoma and prostate cancer, which were increased at baseline. As expected, the serum levels of IgE decreased in both patients after the initiation of dupilumab therapy. IgE plays an important role in the pathogenesis of AD and its concentration is elevated both in serum and on the skin of patients. There is a significant association between higher IgE levels and disease severity [29]. The correlation between high IgE levels and the risk of developing cancer has not been proven [30].

The patients benefited from the effect of dupilumab; they all achieved EASI75, which means an improvement in clinical status in terms of atopic dermatitis of at least 75% compared to baseline, as well as relief from intense pruritus and a significant improvement in their quality of life.

We observed an absence of recurrence or progression of malignant disease, and no patient to date developed a second malignancy. No adverse events related to the dupilumab treatment were observed. All patients handle the subcutaneous application of dupilumab on their own in the home setting, and they all come for regular visits to our department and keep in touch with their oncologist.

Considering the lack of direct association between IL-13 and IL-4 blockage and cancer development, and considering the absence of cancer recurrence, there is no evidence to exclude dupilumab for atopic dermatitis patients with a previous diagnosis of malignancy. Of note, dupilumab is not considered immunosuppressive, but rather immunomodulatory, given its targeted action on the immune system [31]. In atopic dermatitis, the efficacy of IL-13 and IL-4 inhibition has gradually accumulated, and real-world evidence of dupilumab treatment has confirmed its high effectiveness [13,16,17,18,23,31,32]. In 2022, we published a multicenter prospective real-life experience study in the treatment of atopic dermatitis with dupilumab, as well as its effectiveness and safety. As expected, dupilumab showed very good efficacy and was well tolerated. The effectiveness of dupilumab was expressed by a significant reduction in EASI and DLQI scores. After 4 months, EASI75 response was seen in 66.6% of the patients, which further increased after 1 year to 89.5% [23].

In our patients, continuing therapy with dupilumab did not show cancer progression or recurrence, which suggests that dupilumab can be a safe treatment for atopic dermatitis in patients with advanced cancer. It should be noted that it is always important to evaluate each patient individually since there is still a lack of clinical data regarding the use of dupilumab in patients with atopic dermatitis and malignancy.

Clinicians must be cautious and consider treatment on a case-by-case basis following discussion with an oncologist. Further studies with a higher number of patients and longer follow-up are needed.

## Figures and Tables

**Figure 1 jcm-12-01191-f001:**
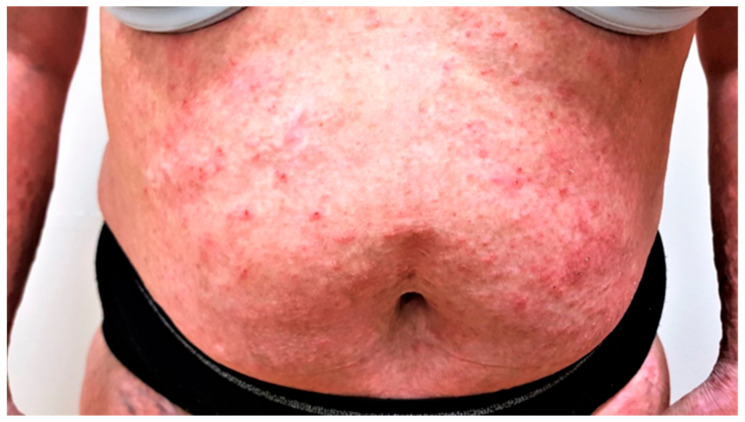
Patient before the initiation of dupilumab with EASI score of 50.

**Figure 2 jcm-12-01191-f002:**
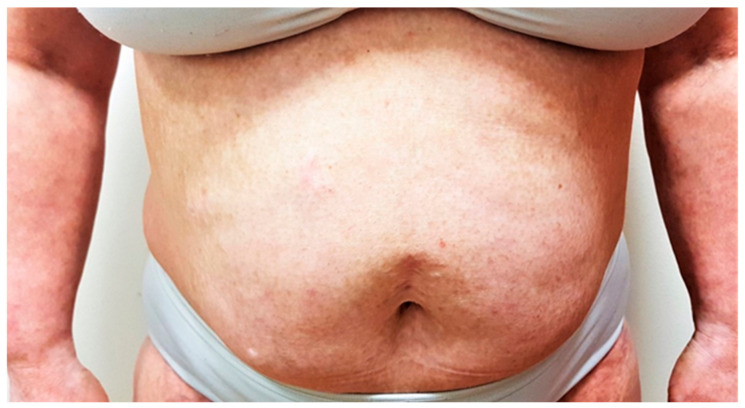
Patient after 2.5 years on dupilumab with EASI score of 8.

**Figure 3 jcm-12-01191-f003:**
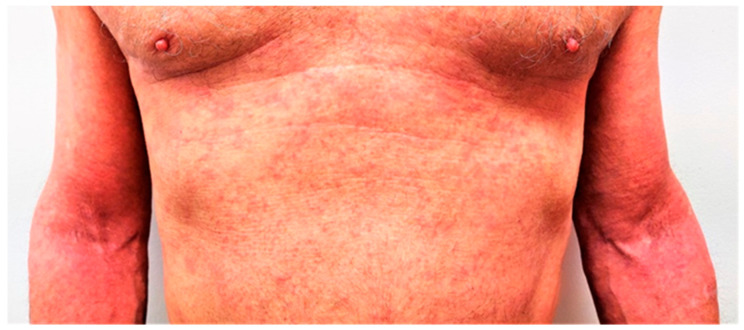
Patient before the initiation of dupilumab with EASI score of 32.9.

**Figure 4 jcm-12-01191-f004:**
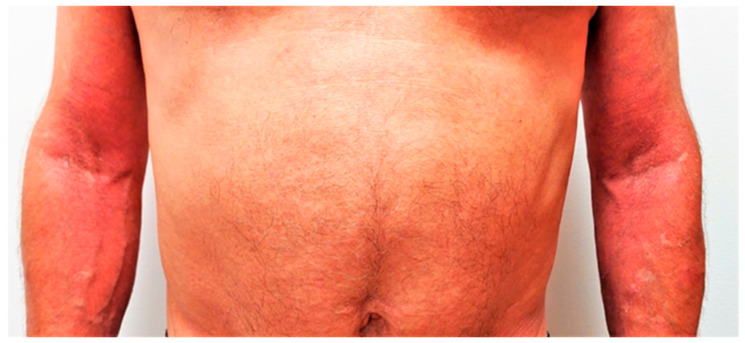
Patient 4 months after the initiation of dupilumab with EASI score of 4.

**Table 1 jcm-12-01191-t001:** Oncologic patients with atopic dermatitis treated with dupilumab.

	Cancer before Dupilumab Initiation	Type of Cancer	Previous Cancer Treatment	Year of Cancer Diagnosis	Year of Dupilumab Initiation	Cancer Recurrence	EASI at the Baseline	Current EASI Score
**Case 1**	Yes	Colorectal carcinoma	Surgery and chemotherapy	2020	2021	No	24.4	0.9
**Case 2**	Yes	Colorectal carcinoma and kidney cancer	Surgery, chemotherapy, and radiotherapy	2018	2020	No	50	8
**Case 3**	Yes	Penile spinocellular carcinoma and prostate cancer	Surgery	2020	2022	No	32.9	4

## Data Availability

Not applicable.

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
