# Peer review of "Treatment of Severe Atopic Dermatitis with Dupilumab in Patients with Advanced Cancer"

_jcm, 2023, doi:10.3390/jcm12031191_

Round 1

Reviewer 1 Report

The paper describes a very small series of oncologic patients who were treated with dupilumab without significant complications. The subject has been treated in the literature. There are many redundances in the discussion: avoid repeating statements and consideration which have been already written in the introduction. 

This is the list of my suggestions to improve the paper:

Abstract

Line 11: Clarify when systemic therapy is necessary (i.e., moderate/severe forms, difficult areas).

Page 2 

Line 67-69:  "AD can be linked to cancer" is too speculative. Delete.

Line 93: Why do not mention tralokinumab, the second biologic which has been licensed for AD?

Page 3

Line 112: In Czech EASI 21.1. Why do you reference such value to EMA?

Line 112-114: The term Eczema Atopic Score Index must be written without abbreviation at the first mention (line 112) not at the second (line 113).

Line 121-122: Usually parasitic infections are not excluded with investigations before dupilumab. Therefore, you should change with "to treat pre-existing infections”.

Line 127: Bronchiale change to bronchial.

Line 126: Delete, unnecessary.

Line 132: There is no evidence about the statement on metotrexate.

Case series

Patient 1:  You should describe the clinical phenotype of this patient (classic? or other generalized, prurigo ecc).

Figures

Insert EASI before and after if available.

Discussion

There are a lot of redundances and repetitions that you should delete.

Line 240-243

Line 244-246

Line 278-281: You repeated the cancers of these patient. 

Line 289-292: Do not repeat the treatments.

Line 294-298

Line 306: It is not necessary to write about coronary disease. Delete.

Line 322-330: Delete repetitions about IL4 and L13.

Line 335-342: Delete, not pertinent.

References

I did not found references which have treated this subject, in particular.

Owji, J Allerg Clin Immunol 2022.

Mitroi, Life 2002.

Patruno, J Dermatol Treat 2022.

Author Response

Thank you for your important comments.

Abstract

Line 11: Clarify when systemic therapy is necessary (i.e., moderate/severe forms, difficult areas).

Response: Corrected in the text.

Page 2 

Line 67-69:  "AD can be linked to cancer" is too speculative. Delete.

Response: Deleted in the text.

Line 93: Why do not mention tralokinumab, the second biologic which has been licensed for AD?

Response: Your comment is certainly correct. However, it is not available in Czech republic.

Page 3

Line 112: In Czech EASI 21.1. Why do you reference such value to EMA?

Response: We have deleted this part of text to shorten the manuscript.

Line 112-114: The term Eczema Atopic Score Index must be written without abbreviation at the first mention (line 112) not at the second (line 113).

Response: Corrected in the text.

Line 121-122: Usually parasitic infections are not excluded with investigations before dupilumab. Therefore, you should change with "to treat pre-existing infections”.

Response: corrected in the text

Line 127: Bronchiale change to bronchial.

Response: Corrected in the text.

Line 126: Delete, unnecessary.

Response: Corrected in the text.

Line 132: There is no evidence about the statement on metotrexate.

Response: Corrected in the text. However, we always try to avoid all possible immunosupressive drugs in patients with malignant history and if possible we offer the patient modern biological treatment. Every case we discuss with an oncologist.

Case series

Patient 1:  You should describe the clinical phenotype of this patient (classic? or other generalized, prurigo ecc).

Response: Corrected in the text.

Figures

Insert EASI before and after if available.

Response: Corrected in the text of figures

Discussion

There are a lot of redundances and repetitions that you should delete.

Line 240-243 Response: Corrected in the text.

Line 244-246 Response: Corrected in the text.

Line 278-281: You repeated the cancers of these patient. Response: Corrected in the text.

Line 289-292: Do not repeat the treatments. Response: Corrected in the text.

Line 294-298 Response: Corrected in the text.

Line 306: It is not necessary to write about coronary disease. Delete.

Response: Corrected in the text.

Line 322-330: Delete repetitions about IL4 and L13. Response: Corrected in the text.

Line 335-342: Delete, not pertinent. Response: Corrected in the text.

References

I did not found references which have treated this subject, in particular.

Owji, J Allerg Clin Immunol 2022. Reponse: we have used this reference – reference no.26

Mitroi, Life 2002. Reponse:  this reference is not in our article and we couldnt find it online

Patruno, J Dermatol Treat 2022. Reponse: there is no such reference in our article, we have found an article on pubmed by Fargnoli et al (including Patruno C) and we added it to our references as no.32

Reviewer 2 Report

This case series is overall well written and appropriately discusses the option of dupilumab as a suitable option for oncologic patients. I do question the novelty or significance of this idea as in the US we commonly use dupilumab as a safe immunomodulatory agent for patients with contraindications to conventional therapy including but not limited to malignancies. Given the limited number of reports it is worth contributing to the literature. 

Regarding the use of methotrexate my colleagues and I generally consider this to be a relatively safe option for patients with suspected or confirmed malignancy (in contrast to cyclosporine, azathioprine, or mycophenolate for example). 

I would like to request some minor clarifications:

Lines 67-69: "AD can be linked to cancer for a number of reasons, including...low self-esteem and lack of sleep". Is is true that low self-esteem is a risk factor for cancer? If not, I would remove this claim. 

Line 85: Would change "often laboratory controls" to "frequent laboratory monitoring".

Lines 113-117: May consider removing EASI definition as the Introduction section is overall quite long. 

Lines 230-239: Would move this info from discussion to introduction or strongly consider removing it entirely in the interest of space constraints.

Author Response

Thank you for your important comments. In. terms of methotrexate we prefer not to use it in patients with malignant history, if possible we always chose to use modern biological treatment event as a first line. Each case is discussed with an oncologist.

Lines 67-69: "AD can be linked to cancer for a number of reasons, including...low self-esteem and lack of sleep". Is is true that low self-esteem is a risk factor for cancer? If not, I would remove this claim. 

 Response: Corrected in the text -it was too speculative.

Line 85: Would change "often laboratory controls" to "frequent laboratory monitoring".

 Response: Corrected in the text

Lines 113-117: May consider removing EASI definition as the Introduction section is overall quite long. 

 Response: Corrected in the text

Lines 230-239: Would move this info from discussion to introduction or strongly consider removing it entirely in the interest of space constraints.

Response: Removed entirely from the text

Round 2

Reviewer 1 Report

The Author addressed the reviewer suggestions.